# OpenReview forum: "The Linear Geometry of Moral Choice in LLMs"
_ICLR.cc/2026/Conference — ICLR 2026 Conference Withdrawn Submission_

### Official Review · Reviewer_95aj · 2025-10-29

**Soundness:** 1
**Presentation:** 1
**Contribution:** 1
**Rating:** 0
**Confidence:** 5

**Summary:**

The paper claims to investigate moral framing effects in LLMs. The authors assert that the distinction between "impersonal" (utilitarian) and "personal" (deontic) moral dilemmas, based on 20 trolley-style prompts, can be represented as a single, steerable linear direction in the models' activation space. They find that "reasoning-tuned" (CoT) models show more utilitarian tendencies and that intervening along this "moral axis" can flip the model's binary YES/NO answer.
Despite its interesting premise, the paper is fundamentally undermined by critical methodological omissions that make its central claims non-reproducible, a data-scarce experimental design that cannot support its generalized conclusions, and poor, imprecise writing.

**Strengths:**

- The paper tackles a potentially interesting and timely question: whether moral framing effects in LLMs can be captured as interpretable geometric directions and if this linearity could be used for steering.

**Weaknesses:**

- **Critical Omissions and a Complete Lack of Reproducibility.** The paper is fundamentally non-reproducible. The authors present results for three model families—"DeepSeek," "LLaMA," and "Qwen" (e.g., in Tables 1, 2, and 3). However, the methodology section (including Appendix D) only specifies the model used for DeepSeek ("deepseek-llm-7b-chat").
This is a critical failure of basic scientific reporting. Without this information, the results for two-thirds of the models are meaningless. It is impossible to verify, interpret, or build upon these findings, and this omission alone is grounds for rejection.

- **Extremely Narrow Scope and Unjustified Generalization.** The paper's title and abstract make broad claims about "The Linear Geometry of Moral Choice." However, the entire experimental setup rests on a tiny, artificial dataset of **20 highly stylized trolley-like dilemmas** (10 personal, 10 impersonal).
The paper does not find a "moral axis"; it finds a framing axis that separates two small clusters of prompts based on their linguistic and affective content. The leap from "a linear direction separating 10 trolley prompts from 10 other trolley prompts" to the "geometry of moral choice" is a massive, unsubstantiated generalization that the data cannot support.

- **Weak Conceptual.** The reported correlation between the projection on the “moral axis” and the model’s Yes/No answers is largely tautological. The axis itself is computed as the difference between the average hidden states of impersonal (typically “Yes”) and personal (typically “No”) moral dilemmas, meaning the Yes/No distinction is already embedded in how the direction is defined. When the authors later show that projections along this same axis separate or predict Yes and No responses, they are effectively re-demonstrating the separation used to construct it. Thus, the observed correlation is expected by design rather than an independent or causal finding.

- **Writing and Presentation Quality.** The paper is poorly written, but not just in a stylistic sense; it is methodologically imprecise and poorly organized, which obscures its content and reveals the lack of rigor.
    - **Imprecise and Confusing Exposition:** The paper couples this imprecision with dense, unnecessary formalism. Simple operations (like mean-differencing to find a vector) are described with heavy notation, while truly essential information (like model names) is absent. This creates a false sense of rigor.
   - **Vague Motivation:** Key concepts are introduced without clear, high-level motivation. For instance, the "pooled vs. per-pair axis" is presented without a clear explanation of why one might be preferred, and the "CIP metric" is invoked without a strong intuitive justification for its use over simpler metrics. This forces the reader to guess the authors' intent.

**Questions:**

N/A

---

### Official Review · Reviewer_cmwn · 2025-10-30

**Soundness:** 3
**Presentation:** 1
**Contribution:** 2
**Rating:** 4
**Confidence:** 2

**Summary:**

This work reveals that framing effects in LLM moral judgments—whether harm is presented as impersonal (switch-flipping) or personal (direct contact)—concentrate along a single linear direction in activation space, with reasoning-oriented prompting regimes reducing sensitivity to this distinction and yielding more utilitarian (consequence-maximizing) choices. The authors extract this impersonal-personal axis via pooled hidden-state differences normalized by a geometry-aware metric (CIP), then demonstrate that small additive edits along the axis enable monotonic, calibrated steering of decisions while preserving fluency and topical content.

**Strengths:**

This work chose an excellent scenario, the binary moral dilemmas, where the linear geometry is intrinsically present, and used robust statistical methods to present geometric, behavioral and steering results.

The geometrics compared against and the summaries of metrics examined are varied, supporting the results well.

The steering experiments add practical significance to the results, and are surprisingly good.

**Weaknesses:**

There are some issues with presentation. First, main paper sections are too concise: the sections related to CIP metric, projection methods and alternative geometries lack intuitions and explanations for motivations, therefore take some efforts to digest. Also, result sections lack visualized results and summarized conclusions.

I also tend to think the application of this type work narrow. The trolley problem is a thought experiment where only one dimension of values are tested, therefore the linearity persists; but think real-world dilemmas where there's multiple dimensions of values at play, e.g. fairness vs. utility. Maybe it's a good idea to pursue weaker results than linearity but examine scenarios with decomposable value dimensions.

**Questions:**

Can some sort of generalization to questions with more choices than Yes/No, or questions where the concerned value dimension is more than just personal-impersonal be achieved?

Will the results be as significant if the tested problems are scaled up, or switched to unseen synthetic questions?

Can you try steering the responses (e.g. into being more impersonal) with more scalable approaches such as system prompt modification, so as to improve application value?

---

### Official Review · Reviewer_W19m · 2025-10-31

**Soundness:** 3
**Presentation:** 3
**Contribution:** 3
**Rating:** 6
**Confidence:** 3

**Summary:**

This paper investigates framing effects (i.e., "personal" vs. "impersonal") of moral judgments in Large Language Models upon the hypothesis that these effects concentrate along a single latent (linear) dimension in the activation space of LLMs. In this regard, the authors extract impersonal-personal steering vectors, investigate them, and eventually leverage these vectors via additive steering to bear the models towards specific moral behaviors. By comparing models with the same architecture yet with diverse generation approaches (i.e., direct prompting, chain-of-thought, and fine-tuned chain-of-thought), the authors show that reasoning-tuned variants tend to be more utilitarian and less framing-sensitive than non-reasoning variants.

**Strengths:**

- This work provides a key bridge between representation engineering and geometry and normative behaviors in LLMs that can pave the way for valuable studies.
- The methodology is sound, as well as properly supported by detailed rationales and robust motivations.
- The finding about a diverse sensitivity (utilitarian vs sensitive) between decoding approaches (direct vs CoT) is interesting and (mostly) consistent among model families.
- The steering is shown to allow for a controlled shift towards the moral direction with no tangible impact on fluency and thematic content, suggesting this approach allows for very precise interventions.
- The manuscript is very well written, properly organized, and with the right amount of information to understand the proposed approach. The Appendix is particularly dense and enriches the main text with relevant details.

**Weaknesses:**

- The entire framework leverages a single latent moral direction, i.e., personal vs. impersonal, restricting its focus to a single, highly specific moral framing (trolley-style dilemmas). This contrasts a bit with the overall title of the manuscript, as while the authors show that this single type of moral framing is linearly separable, this might not generalize to other cases than the trolley-style ones.
- The proposed work is based on only 20 prompt pairs, and the selection of such examples is not precisely motivated. What is the effect of changing this number? What was the rationale for the selection?
- As the applicability of the proposed approach is demanded to the possibility to find an effective steering direction and a proper steering coefficient, some additional insights (also considering the ones provided) into the impact of layers and token positions on the final "steering capabilities" should be provided, such as additional cues on the separability of moral framings across layers.

**Questions:**

- Related to the last weakness: the last hidden layer, also known as "unembedding", might be related more to stylistic separation than conceptual one. In this regard, it would be interesting to see where the moral framings start to separate. Providing some visualization or tables about the separability with respect to layers would be valuable.
- How much is the proposed methodology robust to stylistic shifts? If the separation occurs wrt the "moral axis" rather than a more stylistic one (i.e., personal or impersonal wordings), replacing emotionally loaded verbs with alternatives should not affect too much the identified directions, right?
- More than a comment, it is a suggestion: some relevant works in representation engineering and activation steering seem to be missing (e.g., Arditi et al., NeurIPS 2024). A better literature review would be appropriate.

---

### Official Review · Reviewer_tghn · 2025-10-31

**Soundness:** 3
**Presentation:** 2
**Contribution:** 3
**Rating:** 6
**Confidence:** 2

**Summary:**

This work finds a linear dimension that represents a potentially interesting moral dimension in the final layer of LLM outputs. The authors study this dimension and find that it is robust to prompting method, base LLM, and reasoning mode. They show that this dimension can be used to steer judgments in interpretable ways.

**Strengths:**

As someone with little background in interpretability and a strong background in moral psychology and AI I found the results and research question very interesting. I liked the replications across three different base models and the results in 4.1 are both intuitive and surprising.

**Weaknesses:**

The technical sections of the paper are very hard to understand.

It would be nice to include an example of impersonal/personal in the introduction for those who are not familiar.

Please define the notation in Figure 1. The figure it impossible to interpret without extensive reference to the paper.

More work should be cited in Section 3. It is not clear what is novel to this work and what is building off of well-known metrics and formalisms. It is very difficult to follow or understand the method for a reader who isn’t already working on these methods. I looked at the one paper cited, Park et al. 2024, and it was quite different from the method used here. There are lots of terms that are not described (e.g., decision tail, concept axis, and no steering runs). Likewise, the alternative geometries are just given as jargon, and there is little intuition for what aspects of robustness they get at. The use of lists throughout this section is part of the problem. The variable V is not defined in the notation.

What does this mean: “forward/backward sweeps show negligible hysteresis,
consistent with a smooth one-parameter manipulation,” and what is the data that supports this?

**Questions:**

The main text should say where the stimuli come from. Are these new or have they been published online (and hence part of the training data)? The results need to be replicated on a novel set of stimuli that are not some of the most widely published items in psychology.

Furthermore, the stimuli from Green et al 2001 have been criticized for not being solely about personal or impersonal force. Intention to harm and other confounds in the stimuli should be discussed. See Mikhail, John. "Universal moral grammar: Theory, evidence and the future." Trends in cognitive sciences 11.4 (2007): 143-152.

When you flip the scenarios so that Yes and No end up having the opposite causal effect, do the results remain robust?

It is not clear what tuned-CoT is or how it is generated? Is this a LRM? Please use the same terms throughout.

Were the decision rules actually pre-registered? There is no link to a preregistration statement.

---

### Note · Authors · 2025-12-04

**Comment:**

We respectfully request the withdrawal of this submission so that we can further develop and improve the paper and its core contributions. We appreciate the reviewers time. Thank you!

**Withdrawal Confirmation:**

I have read and agree with the venue's withdrawal policy on behalf of myself and my co-authors.